# The influence of saccades on yaw gaze stabilization in fly flight

**Brock A. Davis, Jean-Michel Mongeau** *

Department of Mechanical Engineering, The Pennsylvania State University, University Park, Pennsylvania, United States of America

* jmmongeau@psu.edu

**Data Availability Statement:** All code and data for this study is publicly available on our lab GitHub site: https://github.com/bmslpsu/Smooth_Saccade_Simulation.

**Funding:** This work was funded by the Department of Mechanical Engineering at Penn State

## Abstract

In a way analogous to human vision, the fruit fly *D. melanogaster* and many other flying insects generate smooth and saccadic movements to stabilize and shift their gaze in flight, respectively. It has been hypothesized that this combination of continuous and discrete movements benefits both flight stability and performance, particularly at high frequencies or speeds. Here we develop a hybrid control system model to explore the effects of saccades on the yaw stabilization reflex of *D. melanogaster*. Inspired from experimental data, the model includes a first order plant, a Proportional-Integral (PI) continuous controller, and a saccadic reset system that fires based on the integrated error of the continuous controller. We explore the gain, delay and switching threshold parameter space to quantify the optimum regions for yaw stability and performance. We show that the addition of saccades to a continuous controller provides benefits to both stability and performance across a range of frequencies. Our model suggests that *Drosophila* operates near its optimal switching threshold for its experimental gain set. We also show that based on experimental data, *D. melanogaster* operates in a region that trades off performance and stability. This trade-off increases flight robustness to compensate for internal perturbations such as wing damage.

## Author summary

Similar to human eyes, flies generate smooth movement and saccades to stabilize and shift their gaze, respectively. Together, these two motor outputs are thought to act synergistically to benefit both visual gaze stability and performance, but their interaction remains poorly understood. To test this hypothesis, we developed a switched (hybrid) control model inspired from flight data of the fruit fly *Drosophila*. Our model supports the notion that a hybrid strategy provides benefits to both stability and performance in flight across a range of visual motion speeds. Our model further suggests that *Drosophila* operates in a region that trades off performance and stability, which could increase flight robustness to internal perturbations such as wing damage.

University. The funders had no role in study design, data collection and analysis, decision to publish, or preparation of the manuscript.

**Competing interests:** The authors have declared that no competing interests exist.

## Introduction

Vision systems in many animals, including humans, combine smooth pursuit movements with ballistic adjustments known as "saccades" [1–6]. This switched or hybrid visual control mechanism works to ensure that stationary and moving targets are centered on the retina [7], as well as minimizing background slip [3, 8]. It has also been proposed that saccade may simplify spatial perception by segregating rotational and translational optic flows [9]. Insects are common subjects of studies of natural vision systems, and like vertebrates, generate smooth and saccadic movement of their head (eyes) and whole body. While much work has been done on the behavioral mechanisms that drive these two types of motion in flying insects [3, 8, 10], understanding their neural basis remains an active field of study [8, 11–13]. Several models have been proposed to capture the nature of flying insect vision control systems, including purely continuous models [11], parallel hybrid models [4], and switched hybrid models [3, 8, 13]. Here, we focus our attention on the latter in the context of yaw gaze stabilization in fly flight.

In the most general sense, a hybrid dynamical system is one that exhibits both continuous smooth dynamics and discontinuous or discrete dynamics. In regions of the state space where continuous dynamics are active, the state transition proceeds smoothly in time according to the differential equations that describe the system. Upon reaching certain state conditions, however, the system transitions to a discrete regime where its behavior is described by a set of discontinuous difference equations.

The mathematical construction of a hybrid system is as follows:

$$\begin{cases} \vec{x}(t) \in \mathcal{C}, & \dot{\vec{x}}(t) = f(\vec{x}) \\ \vec{x}(t) \in \mathcal{D}, & \vec{x}^+(t) = g(\vec{x}) \end{cases} \tag{1}$$

where $\mathcal{C}$ and $\mathcal{D}$ are the flow set and jump set, respectively, $f(\vec{x})$ and $g(\vec{x})$ are the flow map and jump map, respectively, and $\vec{x}$ is the state vector [14, 15]. The state vector transitions smoothly in time according to the flow map when in the flow set, and transitions discontinuously according to the jump map when in the jump set. According to this paradigm, we can consider hybrid systems a superset of continuous and discrete systems by recognizing that setting $\mathcal{D} = \varnothing$ yields a purely continuous system and setting $\mathcal{C} = \varnothing$ yields a purely discrete system.

In classic work on vision, Land proposed and simulated a hybrid vision system combining a smooth proportional-derivative (PD) controller with a zero-order-hold direct feed-through [4]. This resulted in a parallel controller where both continuous and discrete systems were always active. Land showed that a purely smooth controller provided the best performance at low frequencies, but hybrid control provided improved high-frequency compensation [4]. In recent work, Mongeau and Frye developed a switched hybrid system model to describe the gaze stabilization system of the fruit fly, *Drosophila melanogaster* [3]. In their model, the control system integrates object position error and background velocity error for object tracking and gaze stabilization, respectively. When the integrated error reaches a threshold, a saccade is initiated. This integrate-and-fire model recapitulates the actual behavior of the fly, as observed in a tethered flight experiment [3], and captures the calcium dynamics of T3 columnar neurons that evoke bar tracking saccades [13]. Collectively, several studies support that the smooth optomotor and saccadic systems operate in parallel with distinct yet interacting neural circuitry [13, 16–20].

Here we substantially expand on previous work by Land in light of recent experimental and theoretical results [3, 21, 22]. We develop a switched hybrid model of the control system with a plant model for the yaw motion of *D. melanogaster*. Using this model, we explore the

parameter space to determine regions of stability and optimum performance. We then determine in what regimes the hybrid system is superior to the continuous system alone, with the aim of gaining a better understanding of the advantages behind the use of saccades in both biological and artificial vision systems.

## Materials and methods

### Model

We developed a hybrid controller model (Fig 1a) for angular velocity compensation (gaze stabilization) about the yaw axis inspired by *D. melanogaster* [3, 21]. The model consists of a continuous feedback loop combined with a first-order plant and a saccadic switching system that is implemented to supervise the transition between the smooth and discrete controllers. This switching system tracks the integrated error between the set point and response. When the integrated error reaches a defined switching threshold ($\sigma$), the continuous feedback controller is inhibited—analogous to an efference copy—and a saccade is executed. This transition also resets the integrator and plant, essentially re-basing the system to a new point (Fig 1b), as hypothesized in flies [1]. For tractability, we do not model the body rotation associated with saccades, but only the interruption in continuous visual feedback during saccades associated with an efference copy [23].

The plant is a first-order transfer function with inertia and damping. The yaw inertia and damping constants are based on previous modeling work in *D. melanogaster* and provide a

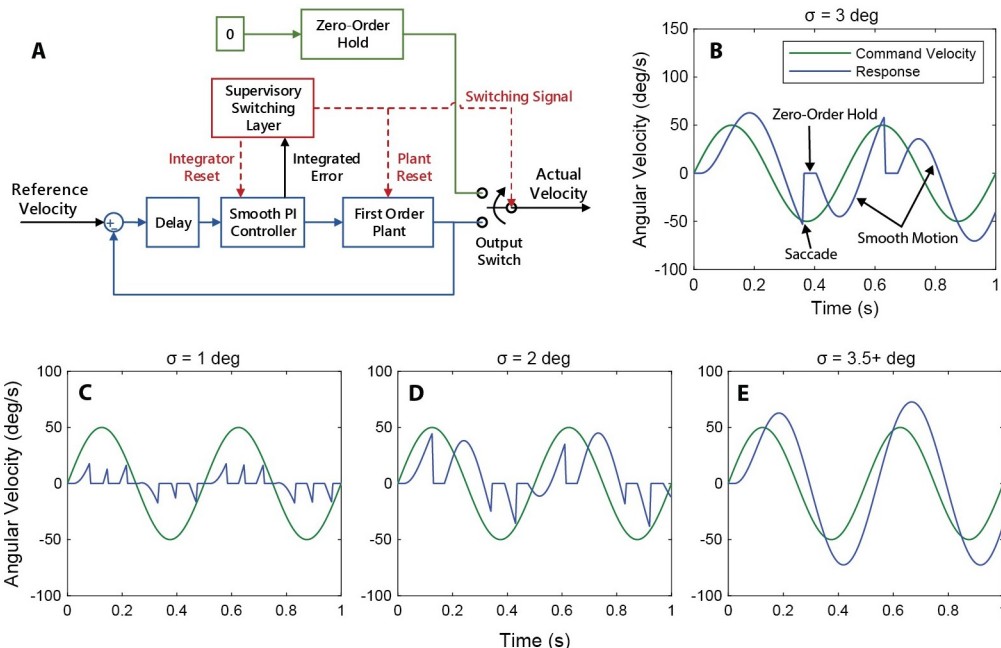

**Fig 1. Design and response of hybrid flight saccade model.** (a) Block diagram of hybrid system model. The hybrid system switches between a smooth PI feedback controller and a saccadic reset based on the integrated error of the smooth controller. (b) Typical response of the hybrid controller to a sinusoidal input velocity. Smooth motion is interspersed with saccades. (c), (d), (e) Evolution of system response as a function of increasing switching threshold, $\sigma$. As the switching threshold is increased, the influence of the saccadic system is reduced. For this particular system ($K_P$, $K_I$, $t_d$) the smooth behavior is recaptured at $\sigma \geq 3.5$ degrees. Note that the gains used in this specific model are not representative of *D. melanogaster*, but are chosen for demonstration only.

damping to inertia ratio of approximately 4.2 [21, 22, 24]. The model was implemented using MATLAB and Simulink.

Within the framework presented in (1), the smooth controller is the flow map, the saccadic reset is the jump map, and the flow and jump sets $\mathcal{C}$ and $\mathcal{D}$ are defined by the switching threshold. We can collapse the hybrid system to be either fully continuous or fully discrete by adjusting the switching threshold towards positive infinity or towards zero:

$$\begin{cases} \sigma \to 0 \Rightarrow \mathcal{C} \to \varnothing \\ \sigma \to \infty \Rightarrow \mathcal{D} \to \varnothing \end{cases} \tag{2}$$

We can intuitively understand this by examining the logic of the switching supervisor. As the switching threshold approaches zero it is more commonly exceeded and the smooth controller is inhibited more often (Fig 1c). In this case saccades are executed in fast succession with little smooth motion between them. Conversely, as the switching threshold is increased towards infinity, the threshold becomes more difficult to exceed until it is no longer possible to initiate a saccade. In this case the smooth controller acts alone, and the system is purely continuous (Fig 1e).

Both the smooth controller and saccadic controller include a transport delay that encapsulates the biological transport time of the visual signal to the muscle of flies. This transport delay has been shown to be around 20 ms in *D. melanogaster* but is kept as a free parameter in the model to allow for exploration of the delay time parameter space [21]. In contrast, the total visuomotor delay in *Drosophila* is approximately 40 ms, which includes phase lag due to inertia [25].

There are three free parameters that define the controller: the proportional gain of the feedback controller ($K_P$), the integral gain of the feedback controller ($K_I$), and the switching threshold ($\sigma$). For a given input velocity, the response is fully defined by these three controller parameters and the transport delay ($t_d$). Because the hybrid system is non-linear, we cannot explicitly write a transfer function between the input and output. However, if we collapse the hybrid system by increasing $\sigma$ as shown in (2), we can recapture the continuous system, which is linear. The open loop transfer function of the continuous system is:

$$G(s) = \frac{\dot{\theta}_{fly}(s)}{\dot{\theta}_{scene}(s)} \quad = D(s)\left(\frac{K_P s + K_I}{s}\right)\left(\frac{1}{Is + C_u}\right) \\ = D(s)\left(\frac{K_P s + K_I}{Is^2 + C_u s}\right) \tag{3}$$

where $\dot{\theta}_{fly}$ is the angular velocity of the fly, $\dot{\theta}_{scene}$ is the angular velocity of the visual background, $G(s)$ is the open-loop transfer function, $D(s)$ is the transfer function of the delay, $K_P$ and $K_I$ are the proportional and integral gains of the controller, respectively, and $I$ and $C_u$ are the plant inertia and damping, respectively. In this representation $I$ carries units of $Nms^2$, $C_u$ and $K_P$ carry units of $Nms$, and $K_I$ carries units of $Nm$. We can reduce the dimensional content of the transfer function by dividing through by $I$, which yields:

$$G(s) = D(s)\left(\frac{\frac{K_P}{I}s + \frac{K_I}{I}}{s^2 + \frac{C_u}{I}s}\right) = D(s)\left(\frac{K_P^* s + K_I^*}{s^2 + C_u^* s}\right) \tag{4}$$

where here $K_P^*$ and $C_u^*$ carry units of $s^{(-1)}$ and $K_I^*$ carries units of $s^{(-2)}$. To form the closed-loop continuous transfer function, we can approximate the delay as a transfer function using a fifth

order Padé approximant, $P^5(t_d, s)$ and simplify:

$$H(s) = \frac{P^5(t_d, s)(K_P^* s + K_I^*)}{s^2 + C_u^* s + P^5(t_d, s)(K_P^* s + K_I^*)}$$

(5)

where $H(s)$ is the closed-loop transfer function for the continuous system. This allows us to use linear system techniques for evaluating stability and performance of the purely continuous system [26].

## Stability

For the continuous system, the stability is determined by the two gains and the delay time. We examine the effect of the three parameters on stability by evaluating Nyquist plots. This allows for a general understanding of how the different parameters affect the stability of the system. To provide a more comprehensive understanding of the stability of the continuous system, we can treat stability as a Boolean function,

$$S_c = f(K_P^*, K_I^*, t_d) \in \mathbb{B}$$

(6)

where an $S_C$ of 1 indicates a stable transfer function and an $S_C$ of 0 indicates an unstable transfer function. Stability for the continuous system is determined using linear system techniques, specifically by examining the poles of the closed loop transfer function. Sweeping through the parameter space then allows for the construction of a stability field with a determination of stability at each analysis point $(K_P^*, K_I^*, t_d)$.

Because the hybrid system is non-linear, we cannot determine stability by examining the transfer function of the system. However, because the execution of a saccade resets the continuous controller and rebases the system at a new position, the hybrid system is inherently stable. Or, more specifically, the hybrid system restabilizes in finite time if it has a finite switching threshold. The bounding of the system is determined by the interplay of the smooth system and the switching threshold of the saccadic system. To describe the stability of the hybrid system, we state:

$$\begin{aligned} S_H \quad &= f(K_P^*, K_I^*, t_d, \sigma) \in \mathbb{R}^+ \\ &= \max(\dot{\theta}_{out}) - \min(\dot{\theta}_{out}) \end{aligned}$$

(7)

where $\max(\dot{\theta}_{out})$ and $\min(\dot{\theta}_{out})$ are the minimum and maximum values of the output angular velocity, respectively. Here $S_H$ does not represent a Boolean determination of stability like the continuous system. Instead, it represents the absolute bounding of the system response for a given set of control parameters. We also note:

$$\begin{cases} S_C = 0 \Leftrightarrow \lim_{\sigma \to \infty} S_H = \infty \\ S_C = 1 \Leftrightarrow \lim_{\sigma \to \infty} S_H \neq \infty \end{cases}$$

(8)

Increasing the switching threshold towards infinity collapses the hybrid system to the continuous system, so the continuous stability condition holds. Thus, if the continuous system defined by $(K_P^*, K_I^*, t_d)$ is stable, $S_H$ is finite for all $\sigma$.

As with the case of the continuous system, we can evaluate the stability of the hybrid system by sweeping through the parameter space—now with an additional parameter, $\sigma$, and determine the stability at each sampling point. However, in this case the stability takes on a real positive value rather than a Boolean value. The magnitude of this value is the maximum absolute

bounding of the output and represents the level of the instability in the specific hybrid system defined by the parameter set $(K_P^*, K_I^*, t_d, \sigma)$.

## Performance

The performance of the system is a function of the excitation frequency in addition to the controller gains, delay time, and switching threshold. Thus, for the continuous and hybrid systems the performance is:

$$
\begin{cases}
\epsilon_c = f(K_P^*, K_I^*, t_d, f_{in}) \in \mathbb{R}^+ \\
\epsilon_h = f(K_P^*, K_I^*, t_d, \sigma, f_{in}) \in \mathbb{R}^+
\end{cases}
\tag{9}
$$

as defined by the error $\epsilon$. To allow for comparison between the purely continuous system and the hybrid system, we chose to use Sum-Absolute Error (SAE). The SAE calculation is a simple sum of the errors between the input and output of the system across the entire time domain signal:

$$
\epsilon_{SAE} = \sum_{t=0}^{t=t_f} |\dot{\theta}_{setpoint}(t) - \dot{\theta}_{out}(t)|
\tag{10}
$$

where $\epsilon_{SAE}$ is the SAE, $t_f$ is the final time of the signal, and $\dot{\theta}_{setpoint}(t)$ and $\dot{\theta}_{out}(t)$ are the target angular velocity and the actual angular velocity, respectively. SAE is an attractive metric because it requires no assumption of linearity. This means that the hybrid system response requires no pre-processing and can be used in its raw form, improving accuracy.

## Results

### Stability—Continuous system

We first studied the influence of the continuous parameters on stability by exploring the evolution of the Nyquist plots as each of the three parameters $(K_P^*, K_I^*, t_d)$ were increased (Fig 2). As expected, increasing the value of any of the three parameters trends towards instability. Fig 3 shows a more comprehensive view of the stability of the smooth system. In the point cloud plot (Fig 3a), orange points represent unstable system parameter sets, and blue points

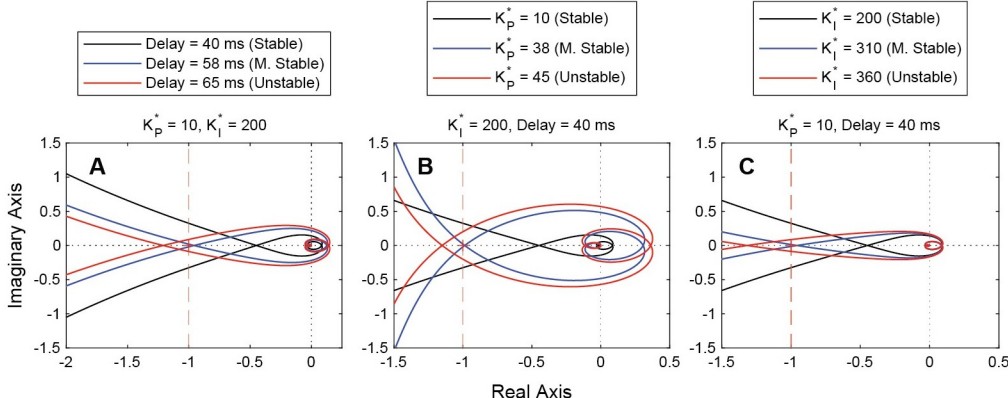

**Fig 2. Nyquist plots for increasing delay and gains.** Increasing any of the three smooth system parameters results in trending towards instability. Vertical dashed line denotes crossover point for stability. (a) Increasing delay from 40 ms to 65 ms with fixed gains. (b) Increasing proportional gain from 10 to 45 with fixed integral gain and delay. (c) Increasing integral gain from 200 to 360 with fixed proportional gain and delay. M. Stable: Marginally stable.

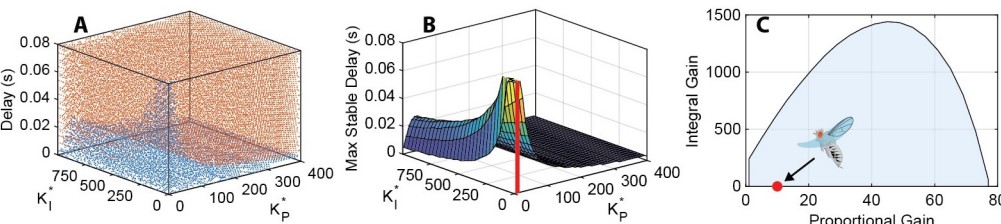

**Fig 3. Stability field for the continuous system.** (a) Binary stability point cloud. Orange points mark unstable gain-delay sets, and blue points mark stable sets. (b) Dividing surface between unstable and stable region. This surface represents the maximum stable delay for a given gain pair. As expected, there is a peak at low gains, implying that such a system can sustain larger delays. There is also a ridge at $K_p^* \cong 35$ for higher integral gains. This $K_p^*$ region has better stability than lower values of $K_p^*$. No gain pair is unstable at delays of zero seconds. The red cylinder represents the experimentally determined $K_p^*, K_I^*$ pair [21]. (c) Slice of surface in (b) at a delay of 20 ms, typical of *D. melanogaster*. The area under the intersection curve represents the region of stable gain pairs. The red point is at the experimentally determined $K_p^*, K_I^*$ pair [21].

represent stable sets. We can then plot the dividing surface, which represents the maximum stable delay for any given gain pair (Fig 3b). Lastly, we can take a slice at the 20 ms delay level, which represents the approximate neural delay of *D. melanogaster*. The stable gain pairs at 20 ms delay are bounded by an approximately parabolic curve. The area beneath the curve represents the gain region of stability for the fly, given the first order model of the plant (Fig 3c). This region matches with experimental system identification models for smooth movement, yaw gaze stabilization in yaw-free flight, which placed the normalized proportional and integral gains at an average of 10 and 1, respectively [21, 22, 25]. These estimates are based on application of an LTI framework through transfer function fitting of experimental gain and phase data from many flies. We previously showed that LTI models captured approximately 90% of the variance in gain and phase of the optomotor response across visual motion frequency.

The surface plot (Fig 3b) generally corroborates the conclusion made from the Nyquist plots —that increasing any of the three system parameters results in instability —but not in every region. There is a ridge in the surface at approximately $K_p^* = 35$ where increasing or decreasing the proportional gain results in instability. The system is at its most stable along this ridge—that is, the system can support longer delay times. In this region, it is lower proportional gains rather than higher ones that result in a more unstable system.

It is also important to note that if there is no delay ($t_d = 0$), there is no set of proportional and integral gains for which the system is unstable. With no delay, the closed loop transfer function is:

$$C(s) = \frac{K_p^* s + K_I^*}{s^2 + \left(C_u^* + K_p^*\right) s + K_I^*} \tag{11}$$

For positive $K_p^*, K_I^*, C_u^*$ the roots of the transfer function are universally stable. This can be seen in (Fig 3b): the surface is elevated above the zero second plane, which implies that all the minimum stable delays are above zero seconds.

## Stability—Hybrid system

We examined the stability of the hybrid controller at the 20 ms delay level by exciting it with a 2 Hz sinusoid with an amplitude of 50 deg/s, which flies can readily stabilize (Fig 4) [21]. As stated previously, adding any finite switching threshold will bound the response of the system

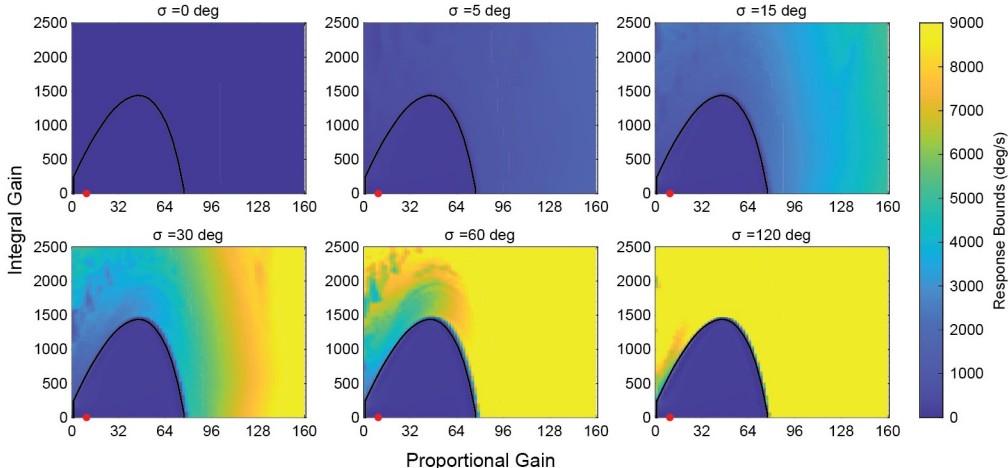

**Fig 4. Effect of switching threshold on stability bounds of the hybrid system.** The black curve is the stability slice for 20 ms—all gain pairs beneath this curve are stable for the continuous system. As the switching threshold is increased, the hybrid system approaches the continuous system; at $\sigma$ = 120 almost all points outside the stability region have bounds above 100 rad/s. Introducing a switching threshold bounds the system response, preventing unbounded instability. This stabilizing aspect of the hybrid system expands the acceptable gain space of the controller beyond what is allowable under continuous control alone. The red point is at the experimentally determined $K_P^*$, $K_I^*$ pair.

at a finite level. We can see that a purely saccadic system ($\sigma = 0$) results in a heavily bounded system response. Increasing the switching threshold reduces the system bounding as $\sigma \rightarrow \infty$, at which point the continuous system behavior is recaptured and instability ensues outside the stable gain region. By introducing the saccadic system of the hybrid controller, the stable gain space can be expanded outside of the bounds of stability for the continuous system.

## Performance

To evaluate the performance of the continuous and hybrid systems at the 20 ms delay level, the system model was excited with pure sinusoids from 0.5 to 3 Hz with amplitudes of 50 deg/s. Simulations were run for hybrid systems with $\sigma$ ranging from 0 to 6 degrees, and the continuous system with $\sigma = \infty$ degrees (Fig 5). Unsurprisingly, as the frequency of the input signal is increased, the error in the response signal increases. However, there are regions in the stable gain space where the increase is less extreme. At higher proportional gains, the error is reduced even at higher frequencies. The optimal region appears to be in high proportional gain $(40 < K_P^*)$ area. This region maintains good performance across all frequencies. The experimental gain pair, marked by the red point, is not within this optimal region. It experienced a dramatic performance reduction as the input frequency is increased.

We next studied the performance of the hybrid system by evaluating the optimal switching threshold for each gain set at each of the sampled frequencies (Fig 6). Shaded regions represent gain sets where the optimal switching threshold was at a level that resulted in a response identical to the smooth system. In these regions the hybrid system provided no performance increase over the smooth system at any switching threshold. At low frequencies, much of the gain space was dominated by regions where the smooth system outperformed the hybrid system. However, as the frequency was increased, the regions where hybrid control provide a performance benefit also increased. The regions with no benefit from hybrid control matched the regions in the gain space where the smooth system maintained good performance—the solid shaded

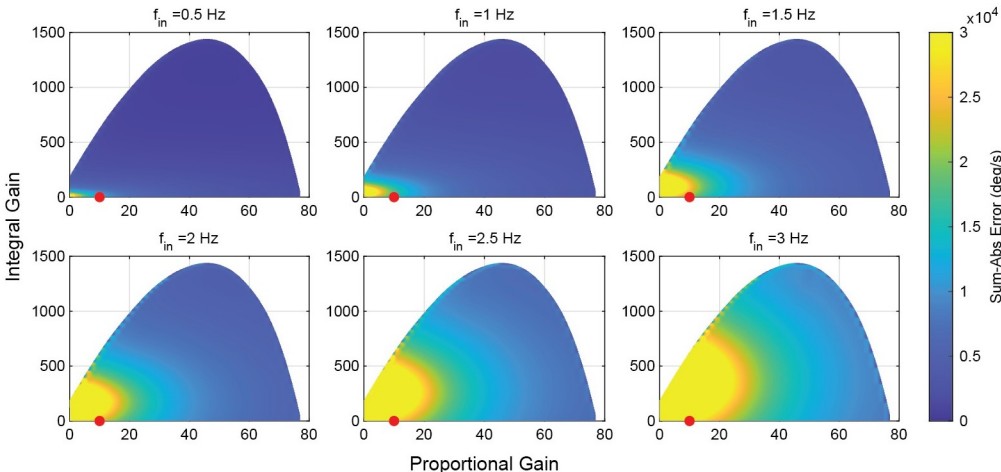

**Fig 5. Error of the purely continuous system as a function of gains and input frequency.** Error increases across the gain space as frequency increases. High proportional gains can act to reduce error, as most low gain systems suffer from a lack of bandwidth at higher frequencies.

areas in Fig 6 aligned with the regions of best performance in Fig 5, and evolved accordingly with the frequency.

The experimentally determined gain set of *D. melanogaster* lies in a region of the gain space that benefits from hybrid control only at higher frequencies, with an optimal switching threshold around 3 degrees. This roughly aligns with experimental work that determined a median saccade initiation integrated error of approximately 6 degrees [3].

Next, we more closely examined the performance of the hybrid controller at three points in the gain space: $(K_P^* = 10, K_I^* = 1)$, the experimentally determined point for *D. melanogaster*

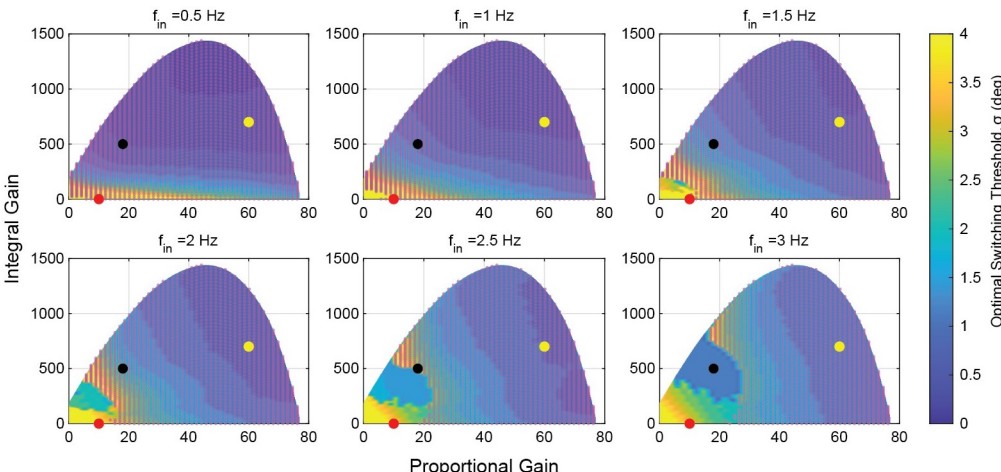

**Fig 6. Optimal switching thresholds at different frequencies across the gain space.** Shaded regions are regions where the optimal switching threshold is at a level that recaptures the continuous system. In these regions the continuous system outperforms any hybrid system. The shaded regions shift as frequency is increased. At the lowest frequencies, only the lowest gain systems benefit from any hybrid control. As the frequency increases, the gain space that is improved by hybrid control increases. At the experimental data point (marked in red), SAE is minimized at higher frequencies with some level of hybrid control, but continuous control is preferred at lower frequencies. The black and yellow points mark the sampling points used in Fig 7b and 7c respectively. Red points: gain pairs where the smooth system outperforms the hybrid system regardless of the switching threshold.

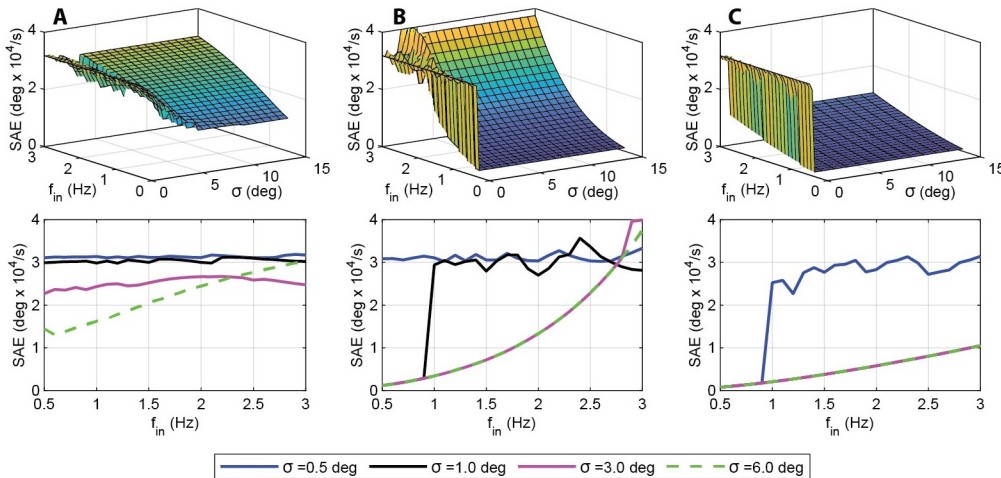

**Fig 7. Error as a function of switching threshold and input frequency for three gain sets.** a) SAE for $(K_P^* = 10, K_I^* = 1)$, the experimental data point (red point in Fig 6). As shown in Fig 6, this gain set lies in a region where hybrid control is preferred at high frequencies. b) SAE for $(K_P^* = 18, K_I^* = 500)$, the black point in Fig 6. This gain set also lies in a region that benefits from hybrid control only at higher frequencies. c) SAE for $(K_P^* = 60, K_I^* = 700)$, the yellow point in Fig 6. This gain set lies in a region that sees no benefit from hybrid control at any of the tested frequencies. Continuous control alone provides the optimal performance. Experimental switching thresholds have been shown to be around 6 degrees [3].

(red point in Fig 6); $(K_P^* = 18, K_I^* = 500)$, the black point in Fig 6; and $(K_P^* = 60, K_I^* = 700)$, the yellow point in Fig 6. These three gain pairs come from distinct regions of the gain space with respect to their hybrid performance. For each gain pair we show the surface plot of the error over frequency and switching threshold in the upper plot, and selected traces in the lower plot (Fig 7).

The first point (experimentally determined) lies in a region where hybrid control enhances performance only at higher frequencies ($f_{in} > 1$ Hz). In Fig 7a there is ridge that splits the hybrid (below $\sigma = 6$ degrees) and continuous systems. This ridge of improved performance runs from 3 degrees to 4.5 degrees and from 1.5 to 3 Hz. In this region the performance is improved by hybrid control, with an optimal switching threshold that varies between 3 and 4.5 degrees. As the switching threshold is increased beyond 4.5 degrees, we can see that the system realigns with the continuous system. At lower frequencies the continuous system outperforms the hybrid system regardless of switching threshold.

The second point lies in a region that also only benefits from hybrid control at higher frequencies. At lower frequencies the continuous system provides the best performance, but at frequencies greater than 2.5 Hz better performance can be achieved with a switching threshold between 0 and approximately 1.5 degrees. The final point lies in a region that does not benefit from hybrid control at any of the sampled frequencies. This point lies in the high proportional and low integral gain region that provides the best performance for the continuous system alone, as shown in Fig 5.

## Discussion

### Model predictions for *D. melanogaster*

Our results show that for the experimentally determined gain set at $(K_P^* = 10, K_I^* = 1)$, saccades provide a performance benefit at frequencies above approximately 1.5 Hz, albeit at

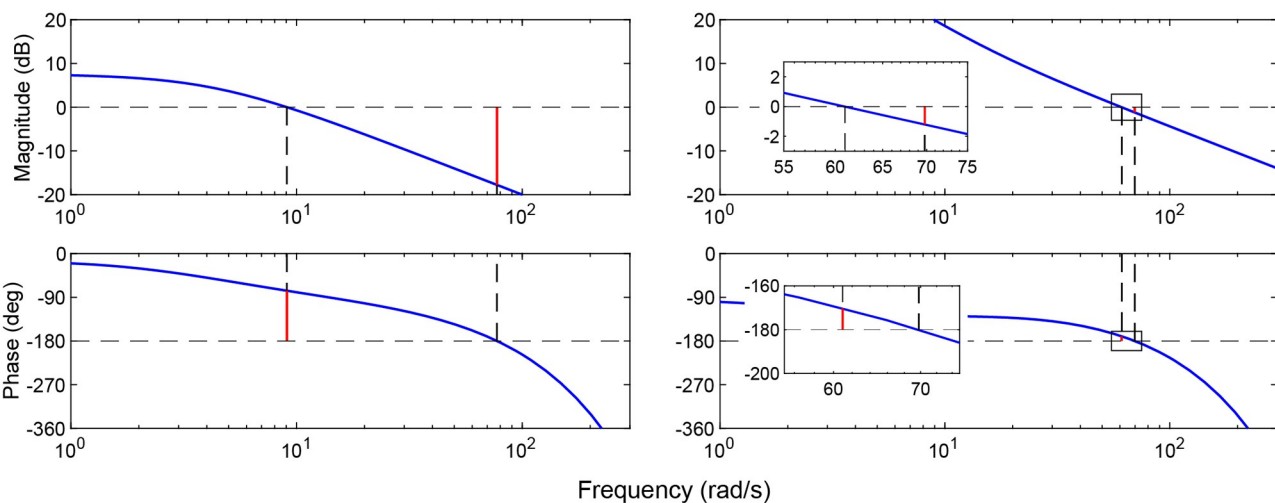

**Fig 8. Gain and phase margins for a model of *D. melanogaster*.** ($K_P^* = 10, K_I^* = 1$) (left) and the ($K_P^* = 60, K_I^* = 700$) system (right). Margins are denoted by red lines and crossover frequencies are denoted by black dashed lines. The modeled fly system has gain and phase margins of 7.8 dB and 104.1 deg, respectively. The high gain system has gain and phase margins of 1.1 dB and 9.8 degrees. Insets show detail of the gain and phase crossover points for the high gain system. This figure simulates the linear system according to Eq (5).

varying switching thresholds. In particular, the optimal switching threshold changes from 4.5 degrees at 1.5 Hz to 3 degrees at 3 Hz (Fig 7). This is close to the 6 degree switching threshold shown in experiments [3]. This implies that *D. melanogaster* operates near the optimal switching threshold for its gain set [3]. This does not imply, however, that the fly operates at the optimal ($K_P^*, K_I^*, \sigma$) point. In the high proportional gain region, the continuous system outperforms the experimentally determined *D. melanogaster* system (Fig 7c).

There are several reasons that may explain why *D. melanogaster* operates outside of the optimal gain region. While increasing proportional gain does result in better performance, it also reduces the stability margins for the closed-loop system. Fig 8 shows Bode plots for the ($K_P^* = 10, K_I^* = 1$) system where *D. melanogaster* operates and the ($K_P^* = 60, K_I^* = 700$) system from Fig 7. The lower gain system has gain and phase margins of 17.7 dB and 103.4 deg, respectively, while the higher gain system has margins of only 1.1 dB and 9.8 degrees. Having larger stability margins would be advantageous in situations where damage occurs to the wings or body of the fly, or aerodynamic conditions change in unexpected ways. The lower gain system is more robust to these types of unanticipated system changes.

Reducing the priorities of any natural system to a single metric such as SAE is inherently a simplification. Performance could include robustness considerations, as mentioned, as well as handling different kinds of input signals, step response performance, and weighing position control priorities as well (as we have only considered velocity control in this work). Furthermore, even if the performance of the biological system could be reduced to a single metric, there is no requirement for a natural system to arrive at an optimal point. It may simply be that there is no evolutionary advantage to improving the compensation system beyond its current state.

Our model predicts that saccade rate could be influenced by the frequency or speed of visual input (and hence visual error). For continuous visual rotation, flies operate with a gain near unity across a range of stimulus rotation speeds ($\approx 100\text{--}500\degree s^{-1}$), but at lower gain at higher speeds [3, 10]. Thus we would expect some influence of stimulus rotation speed on saccade rate, although to our knowledge this not been tested explicitly. Under continuous visual

rotation, stimulus rotation speed influences head saccade rate (which is linked to wing saccades), but these measurements were made in open-loop (body-fixed preparation) which generated large retinal slip due to saturation nonlinearities of the head [8]. Thus, saccades may serve to reset gaze and visual error simultaneously under large accumulated visual error, which could circumvent instabilities in gaze stabilization (e.g. prevent integrator windup).

## Advantages of hybrid control

Though the focus of this paper is on the behavior of *D. melanogaster*, there are several advantages of hybrid control that could be applied to general engineering control systems. The first of these advantages is the capability to expand the stable gain space. Incorporating saccades with a continuous system can allow for stable systems that can run at higher gains than would normally be supported (Fig 4). This could be advantageous in systems that require fast response times but must remain stable after movement such as vision-based tracking systems, machine vision systems, etc. Because saccades create a system that is inherently stable, it could also be used to ensure robustness of stable systems to unexpected conditions.

Hybrid control also provides an advantage for heavily delayed systems by rebasing the system as it drifts away from the setpoint. Though both the saccadic and continuous portions of the system are delayed, the saccades can react much more quickly to changes in the setpoint, especially if the continuous controller has low gain, as is the case for *D. melanogaster*. This provides some compensation for the phase lag introduced in delayed systems by causing the controller to restart from a zero state. This allows for a more responsive controller in situations where system propagation time may be higher than usual (Fig 9).

In addition to reducing delay in the system response, adding saccades to a continuous system also improves performance near resonant frequencies. While the continuous system experiences a large amplitude gain that can push a system close to instability at resonance, the hybrid system keeps the amplitude of the output in check (Fig 9). This could be useful for a system that has good performance outside of a small resonant band. Implementing a hybrid controller may provide for better performance across a larger frequency bandwidth. In general the hybrid controller bonds the amplitude gain of the control system frequency response and decreases the effects of phase lag.

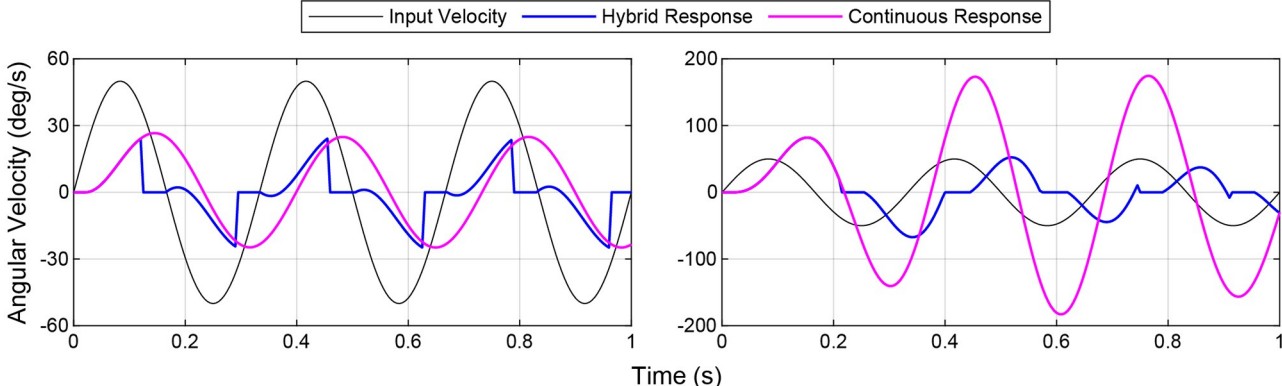

**Fig 9. System input and output for a model of *D. melanogaster* with an adjusted switching threshold.** ($K_P^* = 10, K_I^* = 1, \sigma = 2.8deg, f_{in} = 3Hz$) (left) and a higher gain system ($K_P^* = 10, K_I^* = 500, \sigma = 2.3deg, f_{in} = 3Hz$) (right). The fly model benefits from the addition of saccadic motion from a decrease in the error due to the phase lag. The high gain system is approaching a resonance at this frequency (3 Hz). Introduction of saccadic motion puts bounds on this resonance in a similar way that it puts bounds on an unstable system (See Fig 4), reducing the error and increasing the performance.

## Limitations and future work

There are several key limitations to the model and results presented in this paper, the most important of which is the simplification of the saccadic system. We modeled a saccade as a simple controller and plant reset. In reality, saccades are highly complex, feedback-based mechanisms that propagate through the plant [1, 27]. Future work should include a more detailed model of the saccade dynamics. We also recognize that the $(K_P, K_I, t_d, \sigma, f_{in})$ system space was not completely explored in this work. Specifically, we limited the parameter space by fixing $t_d$ at a fly-relevant 20 ms and choosing a subset of frequencies that might be experienced by *D. melanogaster* in natural situations. Increasing this space to explore the hybrid system performance at other delays and frequencies would be useful in applying results to a broader class of engineering systems and flight regimes. Notably, we used gain parameters in *D. melanogaster* estimated from the yaw optomotor response, but it is known that flies can increase visuomotor gains, e.g. in the presence of an attractive odor [28] or with added inertia [25]. Thus the gain set could be modulated based on context. In addition, we focused on the yaw optomotor response, but in natural flight flies experience complex optic flow that require stabilization across yaw, pitch and roll axes [29]. Lastly, it would be informative to explore performance metrics other than SAE. As mentioned, there are certainly other priorities that inform fly behavior and control. Most notably, incorporating a parallel position controller may provide further insight into the hybrid behavior of vision systems for object-ground discrimination [3].

## Conclusion

Previous work suggested that one advantage of saccades in vision systems is to reduce stabilization error at high frequencies [4]. Here we go substantially further by more fully exploring the parameter space, including mechanics from an experimentally determined plant of *D. melanogaster*, and modeling the hybrid system as switched rather than simultaneous based on recent experimental studies [3, 15, 16, 20]. In doing so, we show that hybrid control does not universally improve performance at high frequencies or speeds. Interestingly, during wide-field, yaw-based gaze stabilization, our model suggests that *Drosophila* operates near its optimal switching threshold for its experimental gain set. However, flies operate at a gain set that does not provide the lowest compensation error. Instead, our work suggests that *Drosophila* operates in a gain region that trades off stability and performance, which could enable robustness to perturbations such as wing damage.

## Acknowledgments

We thank Benjamin Cellini, Bo Cheng and Herschel Pangborn for valuable feedback on the manuscript.

## Author Contributions

**Conceptualization:** Brock A. Davis, Jean-Michel Mongeau.

**Data curation:** Brock A. Davis.

**Formal analysis:** Brock A. Davis.

**Investigation:** Brock A. Davis.

**Methodology:** Brock A. Davis, Jean-Michel Mongeau.

**Software:** Brock A. Davis.

**Supervision:** Jean-Michel Mongeau.

**Visualization:** Brock A. Davis, Jean-Michel Mongeau.

**Writing – original draft:** Brock A. Davis, Jean-Michel Mongeau.

**Writing – review & editing:** Brock A. Davis, Jean-Michel Mongeau.

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
