## [Decision Letter · Decision Letter 0]

1 Nov 2023

Dear Dr Mongeau,

Thank you very much for submitting your manuscript "The influence of saccades on yaw gaze stabilization in fly flight" for consideration at PLOS Computational Biology.

As with all papers reviewed by the journal, your manuscript was reviewed by members of the editorial board and by several independent reviewers. In light of the reviews (below this email), we would like to invite the resubmission of a significantly-revised version that takes into account the reviewers' comments.

We cannot make any decision about publication until we have seen the revised manuscript and your response to the reviewers' comments. Your revised manuscript is also likely to be sent to reviewers for further evaluation.

Sincerely,

Joseph Ayers, PhD

Academic Editor

PLOS Computational Biology

Lyle Graham

Section Editor

PLOS Computational Biology

Reviewer's Responses to Questions

**Comments to the Authors:**

Reviewer #1: This study seeks to extend a model the influence of saccades on yaw gaze stabilization in fly flight. In this research article, the authors extend our understanding of the benefits of combining smooth and saccadic movements in the fruit fly D. melanogaster to improve flight stability and performance.

This study formulates a hybrid control system model that incorporates continuous (smooth) and discrete (saccadic) wing steering movements to stabilize visual gaze against perturbations to optic flow, or the pattern of motion cues projected onto the eyes as the fly flies through a stationary visual landscape. Prior work had done something similar, but without considering the biomechanical constraints of the motor/mechanical locomotory plant. Furthermore, prior work had modeled visual gaze stabilization with, effectively, a simultaneous hybrid controller, whereas here the authors formally consider a switched hybrid controller. The distinction is nontrivial as the two different hybrid control systems involve both continuous and discrete dynamics, but the implications to underlying neural circuit mechanisms are vastly different (and ought to be touched upon in revision).

Prior work by the senior author focusing on saccadic object tracking behavior developed a switched hybrid system that integrates object position error and background velocity error for object tracking and yaw gaze stabilization. Here, the authors more fully explore a model for smooth and saccadic responses that stabilize visual gaze against continuous wide-field optic flow. The model uses state vector transitions according to a flow map, and transitions discontinuously according to a jump map.

The hybrid control system model contains gain, delay, and switching threshold parameters that would contribute to the optomotor yaw stabilization reflex of the fruit fly D. melanogaster. The model includes a first-order plant, a Proportional-Integral (PI) continuous controller, and a saccadic reset system that fires based on the integrated error of the continuous controller. The gain parameter determines the strength of the feedback loop, the delay parameter determines the time delay between the input and output signals, and the switching threshold parameter determines the threshold at which the saccadic reset system is activated.

The study systematically explores the gain, delay, and switching threshold parameter space to quantify the optimum regions for yaw stability and performance. The results show that the addition of saccades to a continuous controller provides benefits to both stability and performance across a range of visual motion speeds. A comprehensive parameter exploration of the model suggests that Drosophila operates near its optimal switching threshold between smooth and saccadic steering movements for its experimental gain set. Based on experimental data, D. melanogaster operates in a region that trades off performance and stability, which increases flight robustness and facilitates compensation for internal perturbations such as wing damage.

The work has several implications for bio-inspired control systems and the design of autonomous flying robots. First, the study shows that the combination of continuous and discrete movements can improve flight stability and performance, particularly at high frequencies or speeds. This finding could inspire the development of bio-inspired control systems that incorporate both types of movements to improve the performance of autonomous flying robots. Second, the study demonstrates the advantages of hybrid control systems, which can expand the stable gain space and provide robustness considerations for handling different kinds of input signals, such as in conditions in which an animal needs to modulate optomotor gain such as when encountering an odor plume. These advantages could be applied to general engineering control systems beyond the behavior of D. melanogaster. Finally, the study highlights the importance of considering trade-offs between performance and stability in the design of bio-inspired control systems and autonomous flying robots. By balancing these factors, it may be possible to improve the robustness and compensation capabilities of these systems. These implications are discussed in more detail by the authors.

There are several limitations to the model and results presented in this study, which are detailed appropriately by the authors. First, the saccadic system was simplified as a simple controller and plant reset. In reality, saccades are highly complex, feedforward and feedback-based phenomena that propagate through the plant. Therefore, future work should include a more detailed model of the saccade dynamics. Second, the parameter space was not completely explored in this study. Specifically, the study limited the parameter space by fixing t_d at a fly-relevant 20 ms and choosing a subset of values for the other parameters. Therefore, further exploration of the parameter space could provide additional insights into the behavior of the hybrid control system. Finally, the study only considered velocity control and did not weigh position control priorities, which would be important when the fly encountered visual objects such as landscape features. These limitations are discussed in detail.

Overall I find the work very interesting, highly rigorous, and timely. This work significantly extends prior models by M.Land (and others) that incorporated relevant visual processing signals, but did not consider constraints based on system biomechanics, nor considered state space transitions. This study will certainly influence future modeling efforts that incorporate underlying neural dynamics. I would endorse publication and have few substantive comments.

Comments:

The authors formalize a switched hybrid controller, which implies parallel yet interacting neural circuitry for the control of smooth optomotor responses and saccadic body reorientations. I would urge the authors to consider the published evidence for the latter both in the Introduction and the Discussion to bolster their case.

An advantage of hybrid control could be discussed is that the biological gain set is modulated by cross-modal behaviors such as odor plume tracking, in which experimental evidence shows that continuous gain is increased, or possibly internal state, in which hunger or thirst modulates search or stability algorithms or heuristics.

Minor:

L49: change to object

L26: change to slip

L56: “motivations” seems odd here, change to “advantages”?

L64: consider changing to: the continuous feedback controller is inhibited by efference copy

L199: change to improved

L82: regarding this citation - how could this include phase lag due to inertia if these flies were rigidly tethered? Maybe you meant to cite a different paper?

Figure 2: define abbreviations in the Legend, such as “M. Stable”

Reviewer #2: Drosophila has been widely used as a model system to study yaw optomotor responses used for stabilizing gaze and forward trajectories in tethered preparations. It has been shown that under these conditions, flies use both smooth following movements as well as saccades to stabilize moving patterns. In this paper, Brock and Mongeau present a hybrid control model for yaw gaze stabilization in Drosophila, which can switch from a continuous proportional-integral control system to a saccadic system, when the error exceeds a treshold. They then explore the parameter space with respect to stability and performance of the system. The hybrid system performs better than a purely continuous system only at high input frequencies and at low gain regimes, in which they claim Drosophila operates. They therefore conclude that flies trade-off performance for stability, which could make the system more robust.

I generally find the paper well written and the findings interesting and a useful contribution to the field. However, I generally think that the paper could be improved by a more in depth discussion as will be discussed in the points below.

1. The conclusion regarding the trade-off between stability and robustness rests on the experimental observation of the relatively low gain parameters, which have been measured in a restricted preparation. How well established are these parameters? How might their estimate be affected by the conditions under which they have been obtained? How variable are they and is it possible that flies can adjust the gains depending on the situation? I think these are questions that should be discussed before drawing any conclusions based on these parameters.

2. Does the model make predictions that could be tested experimentally? E.g. one might expect lower saccade rates at low frequency inputs and higher saccade rates at higher ones. Has something like that be observed? Since optomotor behavior has been so well described, a more in depth discussion of how the model aligns with previous findings would be useful.

Minor:

Fig. 3 legend: Should this be citation 19 here for the gain parameters?

The 3D plots (e.g. in Fig. 7) are hard to interpret. Maybe showing them from a different angle might help, so that the important points are easier to see.

Line 169: It is not immediately clear, what the “shaded regions” are supposed to be in the plot.

Line 199: “improved performance”

Line 212: It is still a bit puzzling to me, why the system would not operate at higher proportional gains. Wouldn’t the risk of instability be mitigated by having a hybrid control system as argued in the paragraph below?

**Have the authors made all data and (if applicable) computational code underlying the findings in their manuscript fully available?**

Reviewer #1: Yes

Reviewer #2: Yes

PLOS authors have the option to publish the peer review history of their article (what does this mean?). If published, this will include your full peer review and any attached files.

Reviewer #1: No

Reviewer #2: No
---

## [Decision Letter · Decision Letter 1]

8 Dec 2023

Dear Dr Mongeau,

We are pleased to inform you that your manuscript 'The influence of saccades on yaw gaze stabilization in fly flight' has been provisionally accepted for publication in PLOS Computational Biology.

Best regards,

Joseph Ayers, PhD

Academic Editor

PLOS Computational Biology

Lyle Graham

Section Editor

PLOS Computational Biology

Reviewer's Responses to Questions

**Comments to the Authors:**

Reviewer #1: I commend the authors on their clear and substantive responses to referees. I find the work novel and compelling publication.

Reviewer #2: The authors satisfactorily addressed my comments.

**Have the authors made all data and (if applicable) computational code underlying the findings in their manuscript fully available?**

Reviewer #1: Yes

Reviewer #2: Yes

PLOS authors have the option to publish the peer review history of their article (what does this mean?). If published, this will include your full peer review and any attached files.

Reviewer #1: No

Reviewer #2: No

---

## [Editor Report · Acceptance letter]

14 Dec 2023

PCOMPBIOL-D-23-01630R1 

The influence of saccades on yaw gaze stabilization in fly flight

Dear Dr Mongeau,

I am pleased to inform you that your manuscript has been formally accepted for publication in PLOS Computational Biology. Your manuscript is now with our production department and you will be notified of the publication date in due course.

With kind regards,

Zsuzsanna Gémesi
